# Non-Targeted Metabolome Analysis with Low-Dose Selenate-Treated *Arabidopsis*

**DOI:** 10.3390/plants14030322

**Published:** 2025-01-22

**Authors:** Hongqiao Li, Tetsuya Mori, Rintaro Moriyama, Moeka Fujita, Genki Hatanaka, Naoki Shiotsuka, Ryota Hosomi, Akiko Maruyama-Nakashita

**Affiliations:** 1Department of Bioscience and Biotechnology, Faculty of Agriculture, Kyushu University, 744, Motooka, Nishi-ku, Fukuoka 819-0395, Japan; li.hongqiao.538@s.kyushu-u.ac.jp (H.L.); moriyama.rintaro.486@s.kyushu-u.ac.jp (R.M.); mfujita@agr.kyushu-u.ac.jp (M.F.);; 2RIKEN Center for Sustainable Resource Science, Yokohama 230-0045, Japan; tetsuya.mori@riken.jp; 3Laboratory of Food and Nutritional Sciences, Faculty of Chemistry, Materials and Bioengineering, Kansai University, Osaka 564-8680, Japan; k299997@kansai-u.ac.jp (G.H.); hryotan@kansai-u.ac.jp (R.H.)

**Keywords:** selenium, selenate, non-targeted metabolome, glucosinolates, flavonoid, amino acids, sulfur metabolism

## Abstract

Selenate, the most common form of selenium (Se) in soil environments, is beneficial for higher plants. Selenate is similar to sulfate in terms of the structure and the manner of assimilation by plants, which involves the reduction of selenate to selenide and the replacement of an S moiety in the organic compounds such as amino acids. The nonspecific incorporation of seleno-amino acids into proteins induce Se toxicity in plants. Selenate alters the plant metabolism, particularly the S metabolism, which is comparable to the responses to S deficiency (−S). However, previous analyses involved high concentrations of selenate, and the effects of lower selenate doses have not been elucidated. In this study, we analyzed the metabolic changes induced by selenate treatment through a non-targeted metabolome analysis and found that 2 µM of selenate decreased the S assimilates and amino acids, and increased the flavonoids, while the glutathione levels were maintained. The results suggest that the decrease in amino acid levels, which is not detected under −S, along with the disruptions in S assimilation, amino acid biosynthesis pathways, and the energy metabolism, present the primary metabolic influences of selenate. These results suggest that selenate targets the energy metabolism and S assimilation first, and induces oxidative stress mitigation, represented by flavonoid accumulation, as a key adaptive response, providing a novel, possible mechanism in plant stress adaptation.

## 1. Introduction

Selenium (Se) is an essential element in various organisms, ranging from bacteria to animals, and is beneficial to higher plants [1,2]. Selenium deficiency causes serious illness, whereas high concentrations of selenium are toxic to humans [2]; hence, cultivation methods and plant breeding strategies are required to both reduce and increase the amount of Se in crops [1,3,4,5,6]. Maintaining Se levels in plants is also crucial to reduce toxicity while preserving its ecological roles, as suggested in recent studies on global Se dynamics [6,7].

In most soil environments, Se exits as selenate. Se and sulfur (S) are in the same group in the periodic table; hence, selenate is structurally similar to sulfate, the form of S available to plants, and is assimilated by plants in the same manner as sulfate [4]. Selenate is taken up by roots via the sulfate transporters (SULTRs) and its assimilation involves the pathway required for sulfate assimilation; this pathway involves the absorption of selenate, its ATP-mediated activation, the reduction to selenite and selenide, and the assimilation into selenocysteine (SeCys). Subsequently, SeCys is converted to selenomethionine (SeMet) and selenoglutathione (SeGSH). The nonspecific incorporation of SeCys and SeMet into proteins induces Se toxicity in plants.

Selenate greatly alters the plant metabolism, particularly the S metabolism. Selenate treatment increases the transcript levels of *SULTRs* associated with sulfate uptake, and decreases the glutathione (GSH) and glucosinolates (GSL) contents [8,9,10,11], which is similar to plant responses to S deficiency (−S) [12]. Transcriptome analysis also revealed a similarity between plant responses to selenate and −S, which is potentially attributed to the competitive interactions between selenate and sulfate during their uptake and assimilation processes [13,14]. Moreover, selenate reduces the accumulation of amino acids, such as proline (about 40% of the control), glutamine (about 50% of the control), asparagine (about 35% of the control), as well as organic acids like isocitrate (about 70% of the control) and α-ketoglutarate (about 40% of the control), which are involved in the TCA cycle [15]. However, these outcomes were reported based on studies using high concentrations of selenate (e.g., 40, 100, and 250 µM) and the effects of lower selenate doses are not reported due to their subtle metabolic changes. Low doses better reflect natural conditions and are critical for exploring the potential benefits of selenate in agriculture and stress tolerance [6,16].

Previous studies have predominantly focused on the metabolic effects of high-dose selenate by employing targeted metabolomic approaches [8,9,10,11,12,13,14,15]. These studies possibly overlooked the subtler impacts of low-dose treatments and limited the discovery of unexpected metabolites. To address these gaps, we aimed to investigate how low-dose selenate impacts the plant metabolism. We hypothesized that low-dose selenate would not inhibit plant growth so much while influencing the primary target of selenate, followed by significant metabolic shifts as adaptive mechanisms to selenate. In the present study, we analyzed the effects of lower doses of selenate on the plant metabolism based on a non-targeted metabolomic approach. We found that low selenate concentrations decreased the S assimilates and amino acids and increased the flavonoids. Additionally, several unknown metabolites were detected, which fluctuated after the selenate treatment.

## 2. Results

### 2.1. High Levels of Selenate Inhibited Plant Growth and Upregulated Low-S-Induced Gene Expressions

We determined the growth conditions suitable for the metabolome analysis of selenate-treated plants using *Arabidopsis thaliana* plants grown on agar medium supplemented with several selenate concentrations (K_2_SeO_4_) (0, 2, 10, 20, and 50 μM) for 2 weeks (Figure 1A). The fresh weight (FW) of the shoot (shoot FW) and root (root FW), and the root-to-shoot ratio were measured to assess the effect of selenate on plant growth. The results showed that selenate suppressed plant growth at all of the tested concentrations. Both shoot and root FWs decreased significantly after the selenate treatment compared to the control counterparts (0 μM of selenate); the growth inhibitory effect was enhanced with an increasing selenate concentration. Moreover, the root-to-shoot ratio was significantly affected by selenate, particularly at 50 μM of selenate.

Considering that selenate at the 2 and 10 μM concentrations exhibited less effect on plant growth compared with higher concentrations, we analyzed the expression levels of the key genes involved in S assimilation, metabolism, and transport (*BGLU28, APR3, SULTR1;1*, and *SULTR1;2*), which are upregulated under −S conditions (Figure 1B). Selenate (10 μM) upregulated the transcript levels of *BGLU28* and *SULTR1;2* in the roots, whereas *APR3* expression was significantly higher after the 2 μM selenate treatment compared to the control counterparts. The expression of these genes was not significant; however, they tended to be induced in the shoots under the 2 and 10 μM selenate treatments. The results indicated that the effects of 10 μM of selenate were similar to those of 2 μM of selenate.

### 2.2. Metabolomic Changes Induced by 2 μM of Selenate in Arabidopsis

As 2 μM of selenate did not severely inhibit the plant growth, but rather sufficiently induced the expression of −S-inducible genes, we conducted a non-targeted metabolome analysis using the plant samples exposed to 0 and 2 μM of selenate (K_2_SeO_4_) to explore the metabolic impact of selenate (Figure 2). The total Se content in soils generally ranges from 0.01 to 2.00 mg kg^−1^, with an average concentration of approximately 0.40 mg kg^−1^ [17]. The 2 µM (0.196 mg kg^−1^) of selenate is considered a low dose, as it falls within the lower range of bioavailable Se levels in soils.

A total of 2515 independent peaks were detected in the samples. The principal component analysis (PCA) exhibited differences between the plants grown under 0 and 2 μM of selenate along the first principal component (PC1) (Figure 2A). Notably, PC1 accounted for 95.8% and 95.9% of the total variance in the positive and negative ion modes, respectively, which indicates that PC1 represents the selenate treatment-induced metabolomic differences. Metabolites that positively and negatively contributed to PC1 were decreased and increased by the selenate treatment, respectively (Figure 2B; Appendix A).

We conducted a metabolic pathway enrichment analysis using the PC1 value with an absolute value of more than 0.54 to understand the metabolic shifts induced by the 2 μM selenate treatment with the KNApSAcK annotation, speculated by the first *m*/*z* (Appendix A), to pick up the metabolites reported in *Arabidopsis thaliana*. Several pathways were enriched, which included amino acid, organic acid, starch and sucrose, and secondary metabolisms, represented by flavone, flavonol, and GSL biosyntheses (Appendix A).

Considering that some of the relevant pathways were modified after the selenate treatment, we further annotated or characterized the detected metabolites following the annotation levels defined by the Metabolomics Standards Initiative (MSI) [14], which included annotation with authentic standards (level 1), the identity with the existing MS/MS data (level 2), or the KNApSAcK database (http://www.knapsackfamily.com/KNApSAcK/ (accessed on 16 December 2024)) using the precursor ion’s *m*/*z* (level 4) (Figure 2B; Appendix A) [19,24]. Amino acids, GSLs, and flavonoids were identified as the three most relevant metabolites affected by the selenate treatment (Figure 2B).

The selenate treatment significantly reduced most of the amino acids, including arginine (Arg), glutamine (Gln), proline (Pro), isoleucine (Ile), phenylalanine (Phe), leucine (Leu), threonine (Thr), valine (Val), pyroglutamic acid, lysine (Lys), and asparagine (Asn) (Figure 2B; Appendix A). Contrastingly, glutamic acid (Glu), aspartic acid (Asp), and tryptophan (Trp) increased, suggesting a potential disruption of nitrogen assimilation or amino acid biosynthesis.

Most of the GSLs, including glucoraphanin, glucoiberin, 4-methylsulfinylbutyl GSL, 5-methylsulfinylbutyl GSL, 7-methylsulfinylbutyl GSL, and glucobrassicin, were decreased by the selenate treatment (Figure 2B; Appendix A); it indicated the downregulation of the GSL biosynthesis or the upregulation of the GSL catabolism. Exceptionally, methoxylated indolic GSL and indolylmethyl GSL + 1MeO increased under the selenate treatment, indicating the selective upregulation of the biosynthesis or the escape from catabolism. Precursors of the indolic GSL pathway, including chorismic acid and tryptophan (Trp), were also increased [9,11].

Flavone and flavonol biosynthesis were also significantly impacted by 2 μM of selenate. The levels of quercetin and kaempferol derivatives, together with the substrates sinapoyl malate, sinapic acid, and 1-*O*-sinapoyl-*ß*-D-glucose, were significantly increased (Figure 2B; Appendix A), suggesting selenate-induced oxidative stress in plants.

### 2.3. Effects of Selenate on the Levels of S and Se-Containing Metabolites

Considering that the GSL levels were highly affected by the selenate treatment, we investigated the impact of 2 μM of selenate on the S metabolism in terms of the total S, the S in the protein fraction, and the levels of sulfate, cysteine (Cys), GSH, GSL, and amino acids in plants (Figure 3).

The total S and S contents in the protein fraction did not differ between the 0 and 2 μM selenate treatment groups (Figure 3A). However, the sulfate content was significantly enhanced (Figure 3B), whereas the phosphate and nitrate levels were severely decreased by selenate (Appendix A). Cys was undetectable in the selenate-treated plants, which is in agreement with the decrease in the amino acid levels observed in the metabolome analysis (Figure 2B; Appendix A). In contrast, the selenate treatment did not affect the GSH levels (Figure 3B).

The levels of GSL, glucoraphanin, glucoiberin, glucohirsutin, and glucobrassicin were severely decreased by the selenate treatment, which is in agreement with the metabolomic analysis results (Figure 2B and Figure 3C; Appendix A). Camalexin is a phytoalexin induced by pathogen attack and abiotic stresses [25,26,27,28,29], which is synthesized from Trp and GSH. Interestingly, we detected a selenate-induced increase in camalexin (Figure 3D; Appendix A).

Unlike the increase in Asp and Glu observed in the metabolome analysis (Figure 2B; Appendix A), all of the amino acids analyzed were decreased by the selenate treatment (Figure 3E and Appendix A). Still, the decrease rate differed among the amino acids, with Asp and Glu decreasing more moderately than in Gln and Pro. Ammonium also decreased under the selenate treatment, suggesting a disruption of nitrogen assimilation.

Moreover, we demonstrated alterations in the total Se level, the Se content in the protein fraction, and the levels of some of the Se-containing metabolites, SeCys, selenite, and selenate (Figure 3F). Both the total Se content and the Se in the protein fraction were detected in significant amounts only in the selenate-treated plants. Approximately 14.12% of the Se existed in the protein fraction. SeCys, selenite, and selenate were detected in the selenate-treated plants, with the highest levels in selenate.

In addition to the S and Se levels, the levels of several other elements were modified by the selenate treatment (Appendix A). The Ca, Mg, Na, and Cu levels were higher in the selenate-treated plants than in the control plants, whereas the Mn, K, Zn, Fe, and Mo levels were decreased.

As some elements associated with photosynthesis, such as Mn, Mg, and Fe, were influenced by selenate, we analyzed the chlorophyll and carotenoid contents. The contents of chlorophyll a and chlorophyll b decreased after the treatment with 2 μM of selenate, while the carotenoid content increased compared to the control counterparts (Appendix A).

## 3. Discussion

Previous studies have demonstrated that selenate can affect the plant growth and metabolism [30,31]. Furthermore, studies have shown that low-dose selenate enhances the S assimilation, antioxidant activity, and secondary metabolite production [32,33]. These effects support Se biofortification strategies, which not only improve the nutritional value of crops, but also enhance their resilience to abiotic stresses, such as drought, salinity, and oxidative damage, aligning with sustainable agricultural practices [1,2,3,4,5]. However, these effects have not yet been clearly elucidated at the metabolic level.

To avoid the side effects of plant growth retardation caused by selenate toxicity, we conducted a non-target metabolome analysis using A. *thaliana* treated with low-level selenate. The results revealed that 2 µM of selenate slightly reduced the plant growth, which was accompanied by many metabolic changes (Figure 1, Figure 2, Figure 3 and Figure 4; Appendix A). Selenate-mediated Cys reduction and the increased expression of −S-responsive genes suggested that the inhibition of S assimilation was the primary selenate-associated metabolic change (Figure 1B and Figure 3B). This may be partially attributable to the competitive assimilation of sulfate and selenate from their uptake via the sulfate transporter *SULTR1;2* into organo-S or -Se metabolites, resulting in an increase in S assimilatory genes [8,34,35,36]. The incorporation of Se into proteins and SeCys was also observed, which is potentially responsible for the reduced plant growth.

Notable metabolic changes derived from the compounds annotated via the non-targeted metabolome analysis concluded that the “decrease in GSL”, “increase in flavonoids”, and “decrease in amino acids” were the major metabolic changes caused by the selenate treatment (Figure 2 and Figure 3). Among these, a decrease in GSL also occurred under −S [37,38]. In addition to the decreased Cys and increased *SULTR* expression, increased sulfate levels may be associated with enhanced sulfate uptake and limited S assimilation (Figure 1B and Figure 3B). Contrastingly, the protein S and total S concentrations remained unchanged, and their amounts in the plants decreased to the extent that FW decreased (Figure 1A and Figure 3A).

Previously, a flavonoid increase under −S conditions was reported [39,40]; however, the selenate treatment exhibited a stronger effect in this study (Figure 2B; Appendix A). The accumulation of flavonoids is a typical response to oxidative stress [41]. The reduced chlorophyll content, which was comparable to previous reports, suggests oxidative stress in plants (Appendix A) [42]. The stability in GSH levels and reduced levels of Cys and GSL suggest that plants prioritize responses against oxidative stress (Figure 3B,C). The increase in the GSH/GSSG ratio detected after the selenate treatment further supports this possibility [30]. Although the GSH levels were unchanged, the increase in flavonoids compensated for the S availability under the selenate treatment (Figure 2 and Figure 3), as flavonoids scavenge ROS effectively and inhibit ROS-generating enzymes, regenerate other antioxidants, and chelate transition metals [43,44,45]. In addition, a trade-off arises from metabolic costs, as selenate-induced flavonoid accumulation, and disrupted nitrogen and S assimilation, divert resources away from primary growth processes to support stress responses.

The overall decrease in amino acid levels differed from the −S response (Figure 2B and Figure 3E; Appendix A). Previous reports showed decreased Cys and increased OAS levels under −S conditions, but the effects on other amino acids differed among the experiments [39,46,47,48]. Treatment of soil-grown *Arabidopsis* with 250 µM of selenate decreased the Asp, Lys, Thr, Ile, Trp, Glu, and Gln contents [15]. Simultaneously, the intermediate organic acids in the TCA cycle, including aconitic acid, citric acid, and 2-oxoglutaric acid (α-ketoglutaric acid), were reduced, suggesting a selenate-mediated inhibition of the energy metabolism [15].

In this study, most of the amino acids were reduced by selenate (Figure 2B and Figure 3E; Appendix A). Although TCA cycle intermediate organic acids were not annotated in this metabolomic analysis, the carbon skeleton of the amino acids was derived from them, suggesting that the energy metabolism was inhibited. The reduction in nitrate, phosphate, and ammonium ions also suggests the inhibition of nitrate and phosphate uptake, and assimilatory nitrate reduction. Nitrate reduction, sulfate reduction, and nutrient uptake require energy and reductants. The relative stabilities of Asp and Glu, which have one amino group, while those of most amino acids, including Asn and Gln, which have two amino groups, decreased (Figure 2B and Figure 3E; Appendix A), which is potentially attributable to the reduced nitrate assimilation compared to the carbon supply from the TCA cycle.

Based on the metabolome analysis, we propose that the primary targets of selenate include S assimilation, energy production, and amino acid biosynthesis. Interestingly, the slight increase in Trp was accompanied by an increase in the Trp-derived secondary metabolites, methoxylated I3M and camalexin (Appendix A). The biosynthesis of methoxylated I3M and camalexin is induced by biotic and abiotic stresses, such as pathogens, AgNO_3_, cold, salt, and UV, which promote plant resistance to insect feed and pathogen attacks [25,26,27,28,29]. In addition to the selenate-mediated promotion of the antioxidant activity [36,49], selenate-associated enhanced disease resistance has been reported in plants [50,51,52]. Although the current treatment with 2 µM of selenate inhibits plant growth, the optimization of the selenate treatment and its concentrations could enhance plant resistance to biotic and abiotic stresses.

In addition to the annotated metabolites, several candidate metabolites were detected through this non-targeted metabolome analysis, which was influenced by the selenate treatment (Appendix A); further research can reveal the corresponding compounds and their physiological functions in plants. Their precise *m*/*z* values, including the fragment ion’s MS/MS spectra provided in this study, make it possible to further explore the selenate-induced metabolic changes in plants.

In conclusion, low concentrations of selenate have potential applications in agriculture, particularly in Se biofortification and enhancing plant stress tolerance. Although the plants in this study were grown on agar media, the optimization of selenate concentrations has potential applications in sustainable agriculture by improving crop tolerance to abiotic and biotic stresses. Low-dose selenate treatment can enhance the antioxidant activity, promote the accumulation of stress-related secondary metabolites, and potentially increase crop resilience to drought, salinity, and pathogen attack. However, achieving this requires careful balancing to minimize the growth reduction and harmful effects on the environment. Future research should focus on determining the crop-specific optimal concentrations and integrating selenate treatments into broader stress management strategies.

## 4. Materials and Methods

### 4.1. Plant Materials and Growth Conditions

The *Arabidopsis thaliana* accession Columbia (Col-0) was used in this study. *Arabidopsis* plants were grown on mineral nutrient (MGRL) agar supplemented with 1% sucrose and 0.5% agarose [53]. In the selenate treatment group, potassium selenate (K_2_SeO_4_) was added to the medium at final concentrations of 0, 2, 10, 20, and 50 μM. The control group was grown under identical conditions without the addition of K_2_SeO_4_. All of the plants were maintained at 22 °C under an 18-h light/6-h dark cycle with a 40 µmol m^−2^ s^−1^ light intensity.

### 4.2. Measurement of Fresh Weight and Preparation of Dry Sample

The shoots and roots were separately harvested from two-week-old plants and rinsed with distilled water. Next, we counted the number of plants, determined their fresh weights (FWs), and stored them at −80 °C. The plant tissues were freeze-dried and stored in a desiccator for the further analyses of the levels of elements, anions, and metabolites.

### 4.3. RNA Isolation and Quantitative RT-PCR

Total RNA was isolated from the shoots and roots using Sepasol-RNA I Super G (Nacalai Tesque, Kyoto, Japan), and reverse-transcribed using the PrimeScript RT Reagent Kit with the gDNA Eraser (Takara, Shiga, Japan). Quantitative PCR was performed on a qTOWER^3^G touch system (Analytik Jena, Jena, Germany) using the KAPA SYBR FAST qPCR Master Mix (2×) (Kapa Biosystems, Boston, MA, USA). Relative mRNA levels were quantified using the ΔΔCt method using actin (*ACT2*) as the internal control. The sequences of the gene-specific primers used for the qPCR are listed in Appendix A.

### 4.4. Non-Targeted Metabolomics

The metabolites were extracted from the powdered dried plant tissues using 80% methanol containing 2.5 µM of lidocaine and 2.5 µM of 10-camphor sulfonic acid, as previously described [19]. The sample was centrifuged, and the supernatant was filtered and analyzed. The extracts were analyzed using LC-QTOF-MS (LC, Waters Acquity UPLC system; MS, Waters Xevo G2 Q-TOF), with previously reported analytical conditions [19].

Data processing for peak picking and alignment was conducted using MS-DIAL ver. 4.70 (http://prime.psc.riken.jp/compms/msdial/main.html (accessed on 16 December 2024)) [54], using previously reported parameters with modifications (minimum peak height: 500; smoothing level: 3). MS-DIAL is an open-source program for MS data analysis that enabled us to reduce missing peak picks and obtain false-positive peaks by optimizing the parameters.

The chemical assignment of the alignment data was performed using MS data from the KNApSAcK database (keyword: *Arabidopsis thaliana*) [24], a database containing information on the metabolites and the species producing those metabolites, from which we can select the metabolites contained in *Arabidopsis thaliana*, Refs. [19,20], and authentic standards. The peaks were manually annotated using MS/MS based on previous reports [19,20] and authentic standards. The annotated metabolites were assigned confidence levels based on the Metabolomics Standards Initiative [18]. A PCA score plot was generated using SIMCA-P 12.0.1 (Umetrics, Umea, Sweden) with Pareto scaling.

### 4.5. Measurement of S and Se in Protein and the Supernatant Fraction

The dried plant samples were ground to prepare a fine powder using a Tissue Lyser II (QIAGEN, Hilden, Germany). The plant powders were extracted with 300 µL of 80% ethanol and stored at 4 °C for 24 h. The mixture was centrifuged at 10,000 rpm for 10 min at 4 °C to precipitate the protein. The supernatant was transferred to a new tube, and the extraction was repeated for the remaining pellet. Supernatants obtained after both extractions were combined and evaporated to dryness. Dried supernatants and protein precipitates were digested in 200 µL of conc. HNO_3_ (Kanto Chemical, Tokyo, Japan) at 90 °C for 30 min, followed by evaporation at 115 °C for 30 min. The digested samples were then diluted to 1.5 mL with ultrapure water and filtered using 0.22 µm filters (TechnoLab, Osaka, Japan). Subsequently, the filtered samples were 10 times diluted using a solution containing 0.1 M HNO_3_ and 10 µg of L^−1^ gallium (Kanto Chemical) as an internal standard. Quantification was performed using a standard curve generated using serial dilutions of standard solutions (Kanto Chemical). Elemental concentrations were analyzed using inductively coupled plasma mass spectrometry (ICP–MS, Agilent, Santa Clara, CA, USA).

### 4.6. Anion Analysis

Plant powder was extracted using 10 mM HCl (Kanto Chemical). The resultant extracts were centrifuged at 4 °C and 13,000× *g* for 15 min, and the supernatants were collected in new tubes. Anion levels were analyzed through ion chromatography (IC-2010; TOSOH, Tokyo, Japan). Anion mixture standard solution 1 (Wako Pure Chemicals, Tokyo, Japan) was considered the standard.

### 4.7. Measurement of Cysteine and Glutathione

The plant extract (refer to Section 4.6) was analyzed; the cysteine (Cys) and glutathione (GSH) contents were then determined using a high-performance liquid chromatography (HPLC) system with fluorescence detection after labeling the thiol groups with monobromobimane, as described previously [38]. The labeled products were separated using a TSKgel ODS-120T column (150 × 4.6 mm, TOSOH) and detected using a fluorescence detector, the FP-920 (JASCO, Tokyo, Japan). The fluorescence of the thiol-bimane adducts was monitored at 478 nm with an excitation wavelength of 390 nm. Cys and GSH (Nacalai Tesque) were used as the standards. The elution was performed using solvents A (12% methanol and 0.25% acetic acid) and B (90% methanol and 0.25% acetic acid) under a gradient program involving 0% to 25% of solvent B.

### 4.8. GSL Measurement

Plant powder was extracted using 200 µL of ice-cold 80% methanol containing 2 µM of l-(+)-10-camphor sulfonic acid (10CS; Tokyo Chemical Industry, Tokyo, Japan) as an internal standard for the negative ionization mode. After homogenization, the cell debris was removed through centrifugation (13,000 rpm for 15 min at 4 °C) and the supernatants were evaporated using a vacuum concentrator. The dried supernatants were dissolved into ultrapure water and filtered using 0.22 µm filters (TechnoLab). The GSL levels were analyzed using an HPLC connected to a triple-quadrupole mass spectrometer (LC-MS8050; Shimadzu, Kyoto, Japan) with an L-column 2 octadecyl-silica (ODS) (pore size, 3 µm, 2.1 × 150 mm; CERI, Tokyo, Japan), as described previously [55].

### 4.9. Amino Acid Analysis

Amino acids were analyzed using the AccQ-Tag system following the instructions provided by the manufacturer (Waters, Milford, MA, USA). The plant extract (refer to Section 4.6) was derivatized with the AccQ-Fluor Reagent Kit (Waters), separated using an AccQ-Tag column (150 × 3.9 mm, Waters) with the custom eluent, and detected using a fluorescence detector, the RF-20A XS (Shimadzu), connected to an HPLC system (Prominence, Shimadzu).

### 4.10. Camalexin Measurement

Dried plant powder was extracted with 100% methanol supplemented with 2 μg mL^−1^ of thiabendazole (Kanto Chemical). After homogenization and centrifugation (13,000 rpm for 15 min), the supernatants were evaporated using a vacuum concentrator. The resultant extracts were dissolved in 80% methanol and analyzed as previously described using an HPLC system (refer to Section 4.9) [56].

### 4.11. Speciation of Se-Containing Metabolites

Plant powder was extracted with 500 µL of 1% HCl, centrifuged (5000 rpm for 6 min at 4 °C), and the supernatants were filtered with 0.2 µm filters. Speciation of Se-containing metabolites were performed using an HPLC system (LC-20Ai; Shimadzu) connected to an ICP-MS (ICP-MS2030; Shimadzu) with a Hamilton PRP-X100 column, as described previously [57].

### 4.12. Element Analysis

The concentrations of Se, S, and other elements in the plant samples were determined using inductively coupled plasma optical emission spectrometry (ICP-OES, Agilent 5800; Agilent Technologies). The samples were prepared and analyzed following a previously described method [58].

### 4.13. Chlorophyll and Carotenoid Analysis

Dried plant tissue powder was extracted with 300 µL of N, N-dimethylformamide (DMF, Nacalai Tesque). The samples were incubated overnight in the dark at 4 °C. After centrifugation at 12,000 rpm for 10 min at 4 °C, the supernatant was transferred into each well of a 96-well plate. The absorbance was measured at 663.8 nm, 646.8 nm, and 480 nm using a Multiscan SkyHigh spectrophotometer (Thermo Fisher Scientific, Waltham, MA, USA). The chlorophyll and carotenoid contents were calculated using the following equations:Chlorophyll a = 12.00 × A663.8 − 3.11 × A646 Chlorophyll b = 20.78 × A646.8 − 4.88 × A663.8Chlorophyll a + b = 17.67 × A646.8 + 7.12 × A663.8Carotenoid = (1000 × A480 − 1.12 × Chlorophyll a − 34.07 × Chlorophyll b)/245

### 4.14. Statistical Analysis

In Figure 1B, a one-way ANOVA followed by Tukey–Kramer tests were used, with significant differences (*p* < 0.05) indicated by distinct letter labels. For Figure 1B, Dunnett’s test was applied using GraphPad Prism 9.4.2. Asterisks represent significant differences, as follows: * 0.1 < *p* < 0.05, ** 0.01 < *p* < 0.05, and *** *p* < 0.01. In Figure 3 and Figure 4, Appendix A, Student’s *t*-tests were performed using Microsoft Excel 2021, considering significant differences, as represented by asterisks, as follows: * 0.1 < *p* < 0.05, ** 0.01 < *p* < 0.05, and *** *p* < 0.01.

## Figures and Tables

**Figure 1 plants-14-00322-f001:**
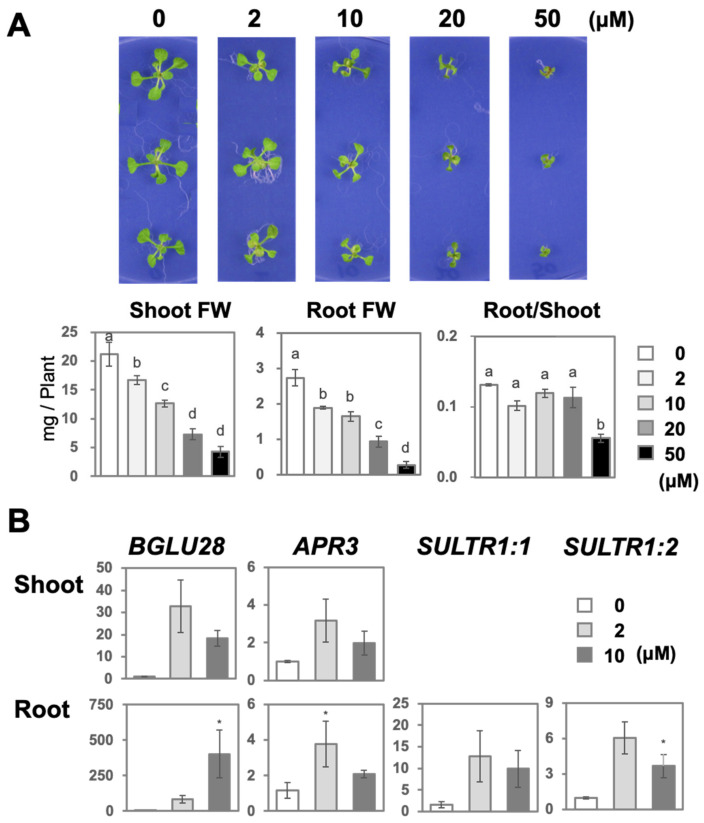
The effects of different concentrations of selenate on plant growth and low-S-inducible gene expressions. Plants were grown for 2 weeks on an agar medium supplemented with various concentrations of K_2_SeO_4_ (0, 2, 10, 20, and 50 μM). RNA was extracted from the plants exposed to 0, 2, and 10 μM of selenate and analyzed via quantitative RT-PCR. (**A**) Representative plant image (top). Shoot (left graph) and root (middle graph) fresh weights, and root-to-shoot ratios (right graph). Bars represent mean ± SE (n = 3). One-way ANOVA followed by the Tukey–Kramer test was performed; significant differences (*p* < 0.05) are indicated by distinct letters. (**B**) Transcript levels of *BGLU28*, *APR3*, *SULTR1;1*, and *SULTR1;2* in the shoots and roots with different treatments. Relative mRNA levels were calculated using the ΔΔCt method, with *ACT2* as an internal control. Bars represent mean ± SE (n = 3). Asterisks denote significant differences compared to the control (0 μM) (Dunnett’s test; * 0.05 ≤ *p* < 0.1).

**Figure 2 plants-14-00322-f002:**
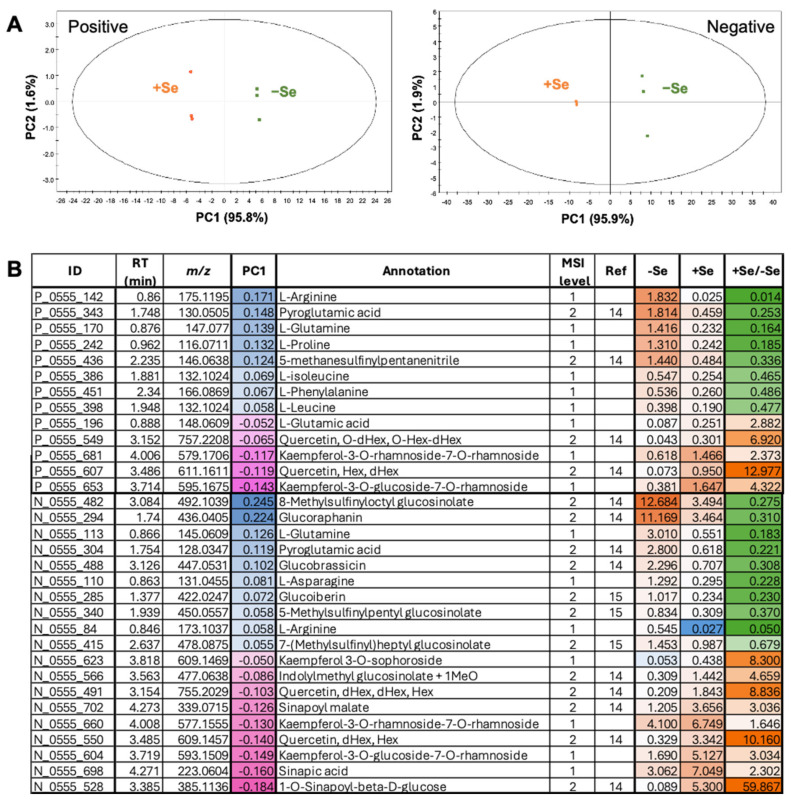
Metabolomic changes caused by selenate. Plants were grown for 2 weeks on agar media supplemented with 0 or 2 μM of selenate (K_2_SeO_4_). After freeze-drying, the samples were subjected to non-targeted metabolome analysis using LCMS. Peak picking and peak annotation to a metabolite were performed as described in the Materials and Methods section. (**A**) PCA of the metabolites detected in the plants grown in the presence of 0 and 2 μM of selenate. (**B**) Metabolites significantly influenced by the 2 μM selenate treatment. We selected the metabolites with PC1 loading values of >0.5 or <−0.5 and curated them for their identities. P and N in the metabolite ID column represent positive and negative ion modes; RT, retention time (min); PC1, loading values for PC1 with the color gradient from magenta to blue representing most minus to most plus values; MSI level [18], metabolites defined by the authentic standard or the MS/MS spectra from the references [19,20,21,22,23]; Ref, reference. −Se, +Se, average of the metabolite intensities when plants were grown 0, 2 µM of selenate with the color gradient from blue to orange representing the lowest to the highest; +Se/−Se, The ratio of intensities between +Se and −Se with the color gradient from green to orange representing the lowest to the highest.

**Figure 3 plants-14-00322-f003:**
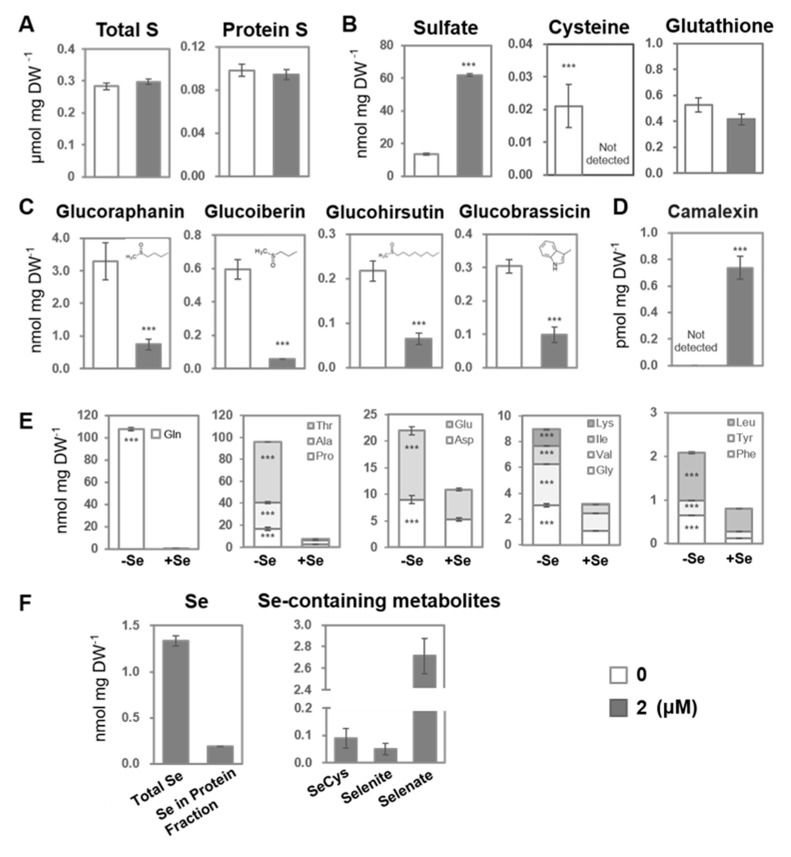
Effects of selenate on the S-, Se-, N-containing metabolite levels in *Arabidopsis thaliana*. Plants were grown for 2 weeks on agar media supplemented with 0 or 2 μM of selenate (K_2_SeO_4_). After freeze-drying, the samples were used for the metabolite analysis, as described in the Materials and Methods section. (**A**) The total S and S contents in protein fractions of the plants grown under 0 or 2 μM of selenate. (**B**) Sulfate, cysteine, and glutathione contents in the plants. (**C**) Glucosinolate contents in the plants. (**D**) Camalexin content in the plants. (**E**) Amino acid content in the plants. (**F**) Total Se, Se contents in the protein fractions, and selenocysteine (SeCys), selenite, and selenate contents in the plants. Bars and error bars represent the mean and standard error (n = 3), respectively. Asterisks indicate significant differences between the two conditions, as determined by Student’s *t*-test (*** *p* < 0.01).

**Figure 4 plants-14-00322-f004:**
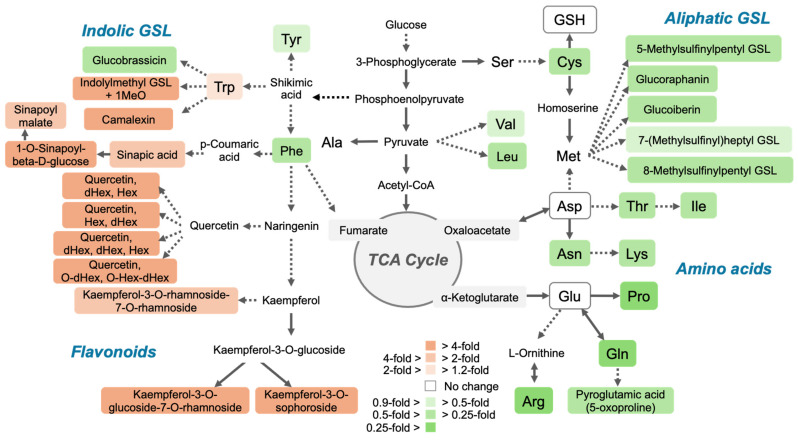
Changes in the metabolite profiles in plants induced by the selenate treatment. Metabolites exhibiting significantly increased and decreased levels are indicated by orange boxes and green boxes, respectively, and the fold changes are indicated by the color gradient, as shown with the boxes on the bottom. Metabolites that were not changed or detected are indicated by open boxes or no background, respectively. Asp and Glu were categorized in the no-change group, as their increase and decrease were not identical between the LCMS and HPLC analyses (Figure 2B and Figure 3E; Appendix A). Continuous arrows represent one-step reactions and dashed arrows indicate a series of biochemical reactions.

## Data Availability

The original data presented in the study are included in the article or Appendix A; further inquiries can be directed to the corresponding author.

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
