# Peer review of "Non-Targeted Metabolome Analysis with Low-Dose Selenate-Treated Arabidopsis"

_plants, 2025, doi:10.3390/plants14030322_

Round 1
Reviewer 1 Report
Comments and Suggestions for Authors
The manuscript entilted "Non-targeted metabolome analysis with selenate-treated Arabidopsis" investigates the metabolic changes induced by low-dose selenate treatment (2 µM) in Arabidopsis thaliana using a non-targeted metabolomics approach. It highlights key metabolic alterations, including reductions in sulfur assimilates, amino acids, and glucosinolates, alongside an increase in flavonoids, suggesting primary disruptions in sulfur metabolism and energy pathways. General Comments
1. Abstract: The abstract is concise but could better describe the broader significance of the results for plant stress physiology.
2. Clarity of Objectives: The research aims are clear, but the introduction could better emphasize why low-dose selenate treatment is understudied compared to high doses.
3. Consistency in Terminology: Ensure consistent use of terms like "selenate" and "selenium treatment" throughout the manuscript.
4. Contextualization of Results: Discussion could be enhanced by comparing the findings with broader implications for selenium biofortification and stress tolerance in plants.
5. Figures and Tables: Some figures lack sufficient detail in captions, especially regarding experimental conditions and statistical significance.
6. Experimental Controls: Clarify if control plants were grown under identical light, temperature, and nutrient conditions without selenate.
7. Relevance of Findings: The study should highlight potential applications of these findings in agriculture or environmental management.
8. Limitations: The discussion lacks an acknowledgment of limitations, such as the use of agar media instead of soil-grown plants.
Specific Comments
1. Title (Line 1): Consider specifying "low-dose" selenate treatment to distinguish this study from others using high doses.
2. Abstract (Line 18): The phrase "primary target of selenate" is vague; specify "sulfur assimilation and amino acid biosynthesis pathways."
3. Abstract (Line 25): The abstract could briefly mention the implications for stress-induced secondary metabolite production.
4. Introduction (Line 33): Add recent references discussing selenium biofortification and its challenges.
5. Introduction (Line 35): Emphasize the ecological relevance of selenium in soil-plant systems.
6. Introduction (Line 48): The statement about amino acid and organic acid reductions under high selenate concentrations could be expanded with specific data.
7. Methods (Line 118): Include details about the agar medium composition to ensure reproducibility.
8. Methods (Line 261): Clarify the choice of 2 µM as the low-dose concentration and whether it reflects natural environmental conditions.
9. Methods (Line 245): Explain the rationale behind the specific metabolomics software and database used.
10. Results (Line 69): The significant effects of 10 µM selenate on gene expression warrant more explanation.
11. Figure 1 (Line 83): Add annotations indicating the specific concentrations tested and statistical significance levels.
12. Figure 2 (Line 102): The PCA plot should include labeled axes with explained variance percentages.
13. Discussion (Line 186): The competitive assimilation between sulfate and selenate is important; cite studies that corroborate this mechanism.
14. Discussion (Line 201): Discuss why flavonoid accumulation is particularly pronounced compared to other stress responses.
15. Figure 3 (Line 171): Highlight the specific metabolites most affected by selenate in a clearer visual format.
16. Conclusion (Line 239): Suggest practical applications of optimizing selenate concentrations for crop stress tolerance.
17. Materials and Methods (Line 250): Include the light intensity and photoperiod duration used during plant growth.
18. Discussion (Line 233): Expand on the potential trade-offs between enhanced stress tolerance and reduced growth in selenate-treated plants.
19. Discussion (Line 226): Include potential future directions for exploring unknown metabolites detected in this study.
Moderate English editing is required.
Author Response
The manuscript entilted "Non-targeted metabolome analysis with selenate-treated Arabidopsis" investigates the metabolic changes induced by low-dose selenate treatment (2 µM) in Arabidopsis thaliana using a non-targeted metabolomics approach. It highlights key metabolic alterations, including reductions in sulfur assimilates, amino acids, and glucosinolates, alongside an increase in flavonoids, suggesting primary disruptions in sulfur metabolism and energy pathways.
General Comments
1. Abstract: The abstract is concise but could better describe the broader significance of the results for plant stress physiology.
Thank you for the helpful comment. We have added a sentence in lines 24-26, “These results suggest selenate would target energy metabolism and S assimilation first and induce oxidative stress mitigation as a key adaptive response, providing a novel possible mechanism in plant stress adaptation.”
Clarity of Objectives: The research aims are clear, but the introduction could better emphasize why low-dose selenate treatment is understudied compared to high doses.
Thank you for the careful comment. We have added descriptions in lines 57-59 with corresponding references. “Low doses better reflect natural conditions and are critical for exploring selenium's potential benefits in agriculture and stress tolerance.”
Consistency in Terminology: Ensure consistent use of terms like "selenate" and "selenium treatment" throughout the manuscript.
Thank you for the careful comment. We have united the descriptions into “selenate treatment” when we describe the treatment. When we describe the results of “selenate treatment,” such as the case “~ was caused by selenate,” we remained “selenate.”
Contextualization of Results: Discussion could be enhanced by comparing the findings with broader implications for selenium biofortification and stress tolerance in plants.
Thank you for the helpful comment. We have described previous studies more in the Discussion, lines 199-204, 231-233, 275-284. We hope this revision will fulfill your request.
Figures and Tables: Some figures lack sufficient detail in captions, especially regarding experimental conditions and statistical significance.
Thank you for the careful comment. We have added experimental conditions and details about the presentations in the legends for Figure 1, line 94, Figure 2, lines 106-109, Figure 3, 187-189.
Experimental Controls: Clarify if control plants were grown under identical light, temperature, and nutrient conditions without selenate.
Thank you for the careful comment. We have added descriptions in the Methods, lines 294-297.
Relevance of Findings: The study should highlight potential applications of these findings in agriculture or environmental management.
Thank you for the helpful comment. We have added a sentence in lines 279-288 to highlight potential applications of our findings in agriculture.
Limitations: The discussion lacks an acknowledgment of limitations, such as the use of agar media instead of soil-grown plants.
Thank you for your helpful comment. We have included the acknowledgment of limitation in the final paragraph in the Discussion, lines 280-283.
Specific Comments
1. Title (Line 1): Consider specifying "low-dose" selenate treatment to distinguish this study from others using high doses.
We have added "low-dose" to the title.
Abstract (Line 18): The phrase "primary target of selenate" is vague; specify "sulfur assimilation and amino acid biosynthesis pathways."
Thank you for the comment. We have specified it as the reviewer suggested in lines 21-24.
Abstract (Line 25): The abstract could briefly mention the implications for stress-induced secondary metabolite production.
Thank you for the comment. We have added the implications to lines 24-26.
Introduction (Line 33): Add recent references discussing selenium biofortification and its challenges.
Thank you for the helpful comment. We have added recent references to line 35.
Introduction (Line 35): Emphasize the ecological relevance of selenium in soil-plant systems.
Thank you for the insightful comment. We tried to emphasize the ecological relevance of Se and the role of plants on that in lines 35-36. However, our study does not contribute to the ecological aspect, so we made it concise. We hope this revision will fulfill your request.
Introduction (Line 48): The statement about amino acid and organic acid reductions under high selenate concentrations could be expanded with specific data.
Thank you for the careful comment. We have specified the reduction rate for amino acids and organic acids in lines 52-55.
Methods (Line 118): Include details about the agar medium composition to ensure reproducibility.
Thank you for the careful comment. We added a reference for the agar medium composition to line 293.
Methods (Line 261): Clarify the choice of 2 µM as the low-dose concentration and whether it reflects natural environmental conditions.
Thank you for the critical comment. The 2 µM selenate is in the range of natural environmental conditions. We added the description to the Results, lines 102-105, to clarify the additional reason for this choice.
Methods (Line 245): Explain the rationale behind the specific metabolomics software and database used.
Thank you for the critical comment. We use MS-DIAL for peak picking and alignment, and KNApSAcK as a database for chemical annotation because we prioritize preventing false positives in data processing. MS-DIAL is an open-source program for mass spectrometry data analysis with many users. We can create a data matrix with fewer missed peak picks and fewer false positive peaks by optimizing the parameters. We adopted the best parameters developed with Prof. Hiroshi Tsugawa, the creator of MS-DIAL. KNApSAcK is a database containing information on metabolites and the plant species producing those metabolites. Using this, we can prevent the annotation of metabolites that do not exist in Arabidopsis thaliana. We explained these in the Methods, lines 319-321, 323-325, and the Results, line 125, to clarify the benefit of using these databases.
- Results (Line 69): The significant effects of 10 µM selenate on gene expression warrant more explanation.
Thank you for the careful comment. We restricted the descriptions to the genes and the treatment; only significant differences were found. To clarify the effects of these two doses, we have added a sentence to lines 85-86.
Figure 1 (Line 83): Add annotations indicating the specific concentrations tested and statistical significance levels.
Thank you for the careful comment. We were sorry that we showed the selenate concentration with square mark on the right of Figure 1B, but the colors differed from the graphs. We have revised the color of the squares in this version.
Figure 2 (Line 102): The PCA plot should include labeled axes with explained variance percentages.
Thank you for the comment. We are afraid that we had labeled PC1 and PC2 axes with percentages explaining the variance in the metabolites. Please let us keep the Figure as it were.
Discussion (Line 186): The competitive assimilation between sulfate and selenate is important; cite studies that corroborate this mechanism.
Thank you for the helpful comment. We added descriptions and references in the Discussion, lines 212-213.
Discussion (Line 201): Discuss why flavonoid accumulation is particularly pronounced compared to other stress responses.
Thank you for the helpful comment. We added a discussion about flavonoid accumulation, which could compensate for the S availability under selenate treatment and its multiple roles to avoid oxidative stress in the Discussion, lines 231-236.
Figure 3 (Line 171): Highlight the specific metabolites most affected by selenate in a clearer visual format.
Thank you for the comment. The most affected metabolites were shown in Figure 2, but the metabolic connection among these metabolites was difficult to interpret, so we mapped the metabolites and the fold changes to the metabolic map. The readers can understand the most influenced metabolites with colors in Figure 3.
Conclusion (Line 239): Suggest practical applications of optimizing selenate concentrations for crop stress tolerance.
Thank you for the helpful comment. We added the application ways to the final paragraph in the Discussion.
Materials and Methods (Line 250): Include the light intensity and photoperiod duration used during plant growth.
We included the light intensity and photoperiod duration in the Methods, line 296-297.
Discussion (Line 233): Expand on the potential trade-offs between enhanced stress tolerance and reduced growth in selenate-treated plants.
Thank you for the helpful comment. We added a discussion about the potential trade-offs in the Discussion, lines 234-236.
Discussion (Line 226): Include potential future directions for exploring unknown metabolites detected in this study.
Thank you for the thoughtful comment. We added descriptions in this part, lines 275-277; however, it relates to our future study using this information, we did not go in depth about the future directions. We hope this revision will be enough.
Reviewer 2 Report
Comments and Suggestions for Authors
The manuscript titled "Non-targeted Metabolome Analysis with Selenate-Treated Arabidopsis thaliana" presents a comprehensive study investigating the impact of selenate treatment on the metabolomic profile of Arabidopsis thaliana. The authors aim to elucidate the subtle metabolic alterations induced by low concentrations of selenate
The research focuses on understanding how 2 µM selenate treatment affects sulfur (S) assimilates, amino acids, and flavonoid levels in Arabidopsis thaliana. Utilizing a non-targeted metabolomic approach, the authors conducted a series of experiments to assess growth parameters, gene expression related to S assimilation, and comprehensive metabolite profiling.
Overall, this manuscript offers a significant contribution to plant metabolomics and selenium research. The authors successfully demonstrate that even low concentrations of selenate can induce substantial metabolic reprogramming in Arabidopsis thaliana. The clear presentation of methods and findings, coupled with the discussion of oxidative stress responses, makes this study a valuable reference for researchers in this field.
I recommend this manuscript for publication.
Author Response
Thank you for the positive comments. We appreciate your time and consideration of the manuscript.
Round 2
Reviewer 1 Report
Comments and Suggestions for Authors
Although the authors have significantly improved the quality of the manuscript in response to the reviewers' comments and suggestions, the last paragraph of the introduction section requires further refinement. It is essential to clearly highlight the research gap by delineating what is already known in the field and what needs further exploration. Additionally, please articulate the aims and objectives of the study, as well as formulate a clear hypothesis. Once these elements are addressed, the manuscript will be suitable for publication.
Comments on the Quality of English LanguageModerate English editing is required.
Author Response
Comment 1
Although the authors have significantly improved the quality of the manuscript in response to the reviewers' comments and suggestions, the last paragraph of the introduction section requires further refinement. It is essential to clearly highlight the research gap by delineating what is already known in the field and what needs further exploration. Additionally, please articulate the aims and objectives of the study, as well as formulate a clear hypothesis. Once these elements are addressed, the manuscript will be suitable for publication.
Response 1
Thank you for the positive comments. We appreciate your additional suggestions for improving the paper's content. We highlighted the research gap between previous and current analysis and clarified the aims of this study in the last paragraph of the Introduction. We hope this revision makes this manuscript suitable for publication in Plants.